# The Role of *BmTMED6* in Female Reproduction in Silkworm, *Bombyx mori*

**DOI:** 10.3390/insects15020103

**Published:** 2024-02-02

**Authors:** Chunyang Wang, Zunmei Hu, Yu Guo, Wenfu Xiao, Youhong Zhang, Anlian Zhou, Ping Chen

**Affiliations:** 1College of Sericulture, Textile and Biomass Sciences, Southwest University, Chongqing 400715, China; cyshine1@163.com (C.W.); 15054952371@163.com (Z.H.); gggguoyu@163.com (Y.G.); 2Sericultural Research Institute, Sichuan Academy of Agricultural Sciences, Nanchong 637000, China; mangzishijia@126.com (W.X.); sczhangyh@126.com (Y.Z.); 18090563232@163.com (A.Z.); 3State Key Laboratory of Silkworm Genome Biology, Southwest University, Chongqing 400715, China

**Keywords:** transmembrane emp24 domain (TMED), silkworm (*Bombyx mori*), female reproduction, dopamine D2-like receptor (Dop2R)

## Abstract

**Simple Summary:**

Transmembrane emp24 domain (TMED) proteins serve as cellular “delivery workers”, facilitating the transport and folding of specific proteins. *TMED6*, a member of the TMED protein γ subfamily, is highly expressed in various tissues in silkworms. In our study, we investigated expression in male and female tissues, as well as in pupal ovaries at different developmental stages. We observe that decreased *TMED6* expression significantly reduces the number of eggs laid by female moths, leading to the accumulation of unlaid eggs in the abdomen. Furthermore, through in vivo and in vitro experiments in silkworms, we have discovered a close relationship between *TMED6* and dopamine D2-like receptors (*Dop2R*). *Dop2R* plays a crucial role in neural signal transmission and can influence reproductive behavior in insects. *BmTMED6* is closely associated with the dopamine receptor, suggesting its potential involvement in regulating the oviposition behavior of the moth through dopamine-related pathways. Our findings will have important implications for sericulture production in the future.

**Abstract:**

Transmembrane emp24 domain (TMED) proteins have been extensively studied in mammalian embryonic development, immune regulation, and signal transduction. However, their role in insects, apart from *Drosophila melanogaster*, remains largely unexplored. Our previous study demonstrated the abundant expression of *BmTMED6* across all stages and tissues of the silkworm. In this study, we investigate the function of *BmTMED6* in reproduction. We observe significant differences in the expression of BmTMED6 between male and female silkworms, particularly in the head and fatboby, during the larval stage. Furthermore, qRT-PCR and WB analysis reveal substantial variation in BmTMED6 levels in the ovaries during pupal development, suggesting a potential association with silkworm female reproduction. We find that reducing *TMED6* expression significantly decreases the number of eggs laid by female moths, leading to an accumulation of unlaid eggs in the abdomen. Moreover, downregulation of *BmTMED6* leads to a decrease in the expression of *BmDop2R1* and *BmDop2R2*, while overexpression of *BmTMED6* in vitro has the opposite effect. These indicate that *BmTMED6* plays a role in oviposition in female moths, potentially through the dopamine signaling pathway. This study provides a new regulatory mechanism for female reproduction in insects.

## 1. Introduction

TMED proteins are a class of type I transmembrane proteins with an approximate molecular weight of 24 kDa. They play an important role in intracellular protein transport and vesicle formation. They are present in the secretory pathways of the endoplasmic reticulum and the Golgi apparatus [1,2]. The TMED family consists of four subfamilies (α, β, γ, and δ), with the number of members varying across different species [3,4]. In mammals, the γ subfamily contains five members [5]. Our previous study suggested that the γ subfamily in insect TMED had four members in Lepidoptera and Diptera, three members in Coleoptera, and two members in Hymenoptera [6]. Insects had fewer members in the γ subfamily compared to mammals, indicating that the evolutionary rate of the γ subfamily in mammals was faster than that in insects. Interestingly, these members of the γ subfamily diverged into three separate subclasses: TMED3-like, TMED5-like, and TMED6-like. Especially Hymenoptera lacked the TMED5-like member, while Lepidoptera and Diptera had multiple TMED3-like members. The orders Lepidoptera, Diptera, Coleoptera, and Hymenoptera all retained the TMED6-like member. The slower evolutionary rate of the γ subfamily in insects compared to mammals, combined with the presence of TMED6-like members in these insect orders, underscored the conserved functional importance of TMED6 in insects [6]. Regarding the function of TMED, there are numerous reports in mammals. The γ subfamily of TMED is associated with various physiological and biological processes, including immune regulation and signal transduction [7,8,9]. It is also implicated in many important diseases, including cancer and diabetes [10,11]. In insects, there are very few reports related to the functional aspects of TMED. Logjam (DmTMED6) is associated with female ovulation behavior, the nuclear factor kappa-B (NF-κB) signaling pathway, and the JUN N-terminal kinases (JNK) signaling pathway [12,13,14]. The silkworm (*B. mori*) is an important economic insect and a well-established model in Lepidoptera [15]. Currently, our team is the only one to have published research on the silkworm *TMED* gene. Our previous experiments indicated that *BmTMED6* was present throughout the entire lifecycle of silkworms [6]. In this study, we demonstrate that the downregulation of *BmTMED6* expression results in a decrease in the number of eggs laid by female moths, and this effect might be mediated by *Dop2R*. These findings provide valuable insights into the molecular mechanisms underlying female reproduction in the silkworm and have important implications for sericulture production in the future.

## 2. Materials and Methods

### 2.1. Silkworm

The silkworm strain Dazao was maintained at Southwest University in China and was cultured/fed (with mulberry) under standard conditions (24–26 °C and 70–85% RH with a photoperiod of 12:12 LD). The *B. mori* wild-type strain Dazao was maintained in our laboratory resource banks at Southwest University (Chongqing, China).

### 2.2. RNA Extraction, cDNA Synthesis, RT-PCR, and qRT-PCR Analysis

Total RNAs were extracted from tissue or cell samples following the manufacturer’s instructions for a total RNA extraction kit (TaKaRa, Beijing, China). The genomic DNA was subsequently eliminated through a 2 min incubation with ezDNase at 37 °C. For the synthesis of the first-strand cDNA, oligo dT primers and Moloney murine leukemia virus (M-MLV) reverse transcriptase (Promega, Beijing, China) were employed. RT-PCR was conducted with an initial denaturation step at 94 °C for 4 min, followed by 28 cycles of denaturation at 94 °C for 40 s, annealing at 58 °C for 50 s, and extension at 72 °C for 50 s per cycle. A final extension step at 72 °C for 10 min concluded the amplification process. Subsequently, qRT-PCR was performed using a 2× SYBR Prime qPCR Mix (Baoguang, Chongqing, China) and executed on an ABI Prism 7000 Sequence Detection System (Thermo Fisher Scientific, Shanghai, China). The *BmTIF4* gene was used as an internal control [16]. Three biological replicates were set for each sample, and the relative expression levels were calculated using the 2^^(−ΔΔCt)^ method [17]. The primers are listed in Appendix A.

### 2.3. Construction of Prokaryotic Expression Vector and Purification of Recombinant Protein

The pET28a/BmTMED6 recombinant plasmid was generated by inserting the emp24 domain sequence, derived from the conserved domain of *BmTMED6*, into the pET-28a(+) vector using *BamHI* and *HindIII* restriction enzymes. The recombinant plasmids were subsequently transformed into viable *E. coli* BL21 cells for in vitro expression. Heterologous expression was induced by the addition of 0.4 mM isopropyl-β-D-thiogalactopyranoside (IPTG) under different temperature and agitation conditions: 16 °C with 180 rpm for 18 h, 28 °C with 200 rpm for 10 h, or 37 °C with 220 rpm for 5 h. The bacterial cells were then harvested by centrifugation at 7155× *g* for 15 min at 4 °C and resuspended in buffer A (0.5 M NaH_2_PO_4_, 0.5 M Na_2_HPO_4_, pH 7.4, 0.5 M NaCl). Cell disruption was achieved using an ultrasonic instrument. The samples were subsequently analyzed by sodium dodecyl sulfate-polyacrylamide gel electrophoresis (SDS-PAGE). The Ni-NTA column (Ni^2+^-nitrilotriacetic acid, 1.6 cm × 20 cm, 10 mL) was equilibrated with buffer A at a flow rate of 2 mL/min. A 0.45 µm filter membrane was used to filter 20 mL of cell lysate dissolved in buffer A, and the samples were introduced at a flow rate of 1 mL/min. The column was washed with buffer A at a flow rate of 2 mL/min for two to five column bed volumes. For stepwise elution, imidazole-containing buffer A at concentrations of 25 mM, 50 mM, 75 mM, 100 mM, 125 mM, 150 mM, 250 mM, and 500 mM was used at a flow rate of 2 mL/min, and elution peaks from each step were collected. The molecular weight and purity of the fusion protein were determined using SDS-PAGE. The ultrafiltration tube (Millipore, Amicon-Ultra-15, Burlington, MA, USA) was cleaned with MilliQ water 2–3 times. It was then centrifuged at 4 °C at 3000× *g* for 20 min. Next, 10 mL of purified protein liquid was added to the ultrafiltration tube, and it was centrifuged at 4 °C, 3000× *g* for 20 min. The liquid in the lower tube was discarded, and centrifugation continued until the protein liquid in the filter tube was reduced to 1–2 mL. The concentrated liquid was collected. This process was repeated until the enrichment was complete. The purified protein, with a concentration exceeding 0.5 mg/mL, was sent to the company (Genecreat, Wuhan, China) for the preparation of polyclonal antibodies. The serum obtained displayed antibody titers exceeding 1:50,000.

### 2.4. Synthesis of Double-Stranded RNA (dsRNA) and Injection

The synthesis of double-stranded RNA (dsRNA) was carried out following the instructions provided in the T7 Ribomax™ Express RNA interference (RNAi) System (Promega, Beijing, China) kit (Appendix A). Cryoanesthetized pupae were rapidly injected into the hemocoel through the third stomatal hole on the right side using a homemade capillary needle containing 5 μL of dsRNA solution (6 ng/μL). Special care was taken to minimize bleeding during the injection process. In the experimental group, we injected 30 newly molted female pupae with dsBmTMED6 synthesized using the above method. In the control group, we injected 30 newly molted female pupae with dsEGFP (the enhanced green fluorescent protein gene that did not exist in silkworms was used as a control) synthesized using the above method. The weight of each female pupa was 0.49 ± 0.03 g. After 24 h of injection, 3 female pupae from the experimental group and 3 female pupae from the control group were randomly selected. After 48 h of injection, 3 female pupae from the experimental group and 3 female pupae from the control group were randomly selected. The same criteria were used to select female pupae at 3, 6, and 9 days and to dissect them to observe ovarian development. The control group and the experimental group were investigated immediately after eclosion. The control group consisted of 15 female moths, and similarly, the experimental group consisted of 15 female moths. They were weighed before mating, and their mating and egg-laying behaviors were observed. In the same batch, 30 untreated male moths mated with two groups of females once and separated four hours later. The female moths laid eggs in the dark in a separate moth circle for 3 days, and we observed the number of eggs. We dissected and analyzed the egg counts of the 15 female moths in the experimental group. In the control group, 15 female moths were dissected for egg count analysis.

### 2.5. Western Blotting Assay

The extracted proteins were separated by SDS-PAGE (12.5% PAGE Gel Fast Preparation Kit, Epizyme, Shanghai, China). The protein concentration was determined using the BCA protein assay, and approximately 20–40 µg of total protein was loaded per well. Following separation, the proteins were transferred onto PVDF membranes using a constant current of 100 mA for 1 h. The membranes were then blocked with 5% dry milk in Tris-buffered saline containing Tween 20 (TBS-T; 10 mM Tris-HCl, pH 7.4, 150 mM NaCl, 0.01% Tween 20) and incubated overnight at 4 °C with primary antibodies. The primary antibodies used were BmTMED6 (diluted 1:3000) or α-tubulin (diluted 1:5000). After incubation, the membranes were washed with TBS-T and further incubated with a secondary antibody conjugated to horseradish peroxidase (diluted 1:5000; Beyotime, Shanghai, China) at room temperature for 2 h. The signals were visualized using an enhanced chemiluminescence detection system (ECL; Thermo Fisher Scientific, Shanghai, China) and captured using a ChemiScope 3400 Mini (Clinx, Shanghai, China). ImageJ (version 1.41) software was used to analyze the gray value of WB bands.

### 2.6. Construction of Overexpression Vector, Cell Culture, and Transfection

The open reading frame sequence of *BmTMED6* was inserted into the pSL1180 vector using *BamHI* and *NotI* restriction enzymes (Appendix A). An empty vector was used as a control. The BmN cell line was cultured at 27 °C in TC-100 insect medium (AmyJet Scientific, Wuhan, China) supplemented with 10% fetal bovine serum. Cell transfection was performed using a Sinofection transfection reagent (Sino Biological, Beijing, China) according to the manufacturer’s instructions. After transfection, the cells were incubated for 24 h for further analysis. The cell line used in this experiment was BmN-SWU1, given by Pan [18].

### 2.7. Statistical Analysis and Data Presentation

All data are reported as means ± SEM (standard error of the mean). One-way analysis of variance (ANOVA) and two-tailed A Student’s *t*-test was employed for conducting multiple comparisons. Multiple comparisons were performed, and a significance level of *p*-values less than 0.05 was considered statistically significant. Highly significant differences were defined as *p*-values less than 0.01 or 0.001. Statistical analysis was conducted using IBM SPSS Statistics 21. Data visualization was performed using GraphPad Prism, version 8.0 (GraphPad Software, San Diego, CA, USA).

## 3. Results

### 3.1. In Vitro Expression and Purification of BmTMED6

The pET-28a(+) vector is used to ligate the non-transmembrane segment of BmTMED6 (amino acid positions 41–211). The resulting recombinant plasmid is induced for expression in the BL21 bacterial strain using 0.4 mM IPTG at three different temperatures. SDS-PAGE analysis reveals that BmTMED6 is predominantly present in inclusion bodies, with the highest concentration observed at 37 °C (Figure 1A). Ni-NTA is employed to separate the recombinant protein containing a His tag, and the recombinant protein is successfully eluted at a concentration of 250 mM imidazole (Figure 1B). The eluted recombinant protein is concentrated using an ultrafiltration tube, resulting in a protein purity of over 95%. The size of the protein is approximately 25 kDa, consistent with the expected size of the fusion protein. Subsequently, the purified protein is used to generate polyclonal anti-BmTMED6 antibodies through collaboration with Genecreat company (Wuhan, China).

### 3.2. Expression of BmTMED6 in Larval Tissues of Females and Males

After accurately distinguishing male and female individuals based on phenotypic differences in larval gonads during the early stage of the 5th instar, larval tissues are collected on the 3rd day of the 5th instar for qRT-PCR analysis. The results demonstrate that *BmTMED6* is expressed in multiple tissues, including the midgut, ovary, testis, silk gland, hemolymph, nerve, and Malpighian tubules, with minimal differences between males and females. However, significant differences are observed in the fatbody and head (0.001 < *p* < 0.01) (Figure 2A). The most important economic traits of silkworms are silk production and reproduction. The fifth instar larval stage is a critical period for silk gland development. Considering the differential expression observed in the fatbody and head, we are conducting WB analysis on these three tissues. The results indicate significant differences in BmTMED6 protein expression between males and females in the fatbody and head, but not in the silk gland (Figure 2B,C).

### 3.3. Expression of BmTMED6 in the Ovary during Pupal Development

*DmTMED6* is a key regulator of oviposition in female *D. melanogaster* [12]. The pupal stage is a critical period for the development and maturation of the reproductive system in female silkworms. It spans approximately 9 days, from pupation to eclosion. Therefore, we examine the expression of *BmTMED6* in the ovary at different time points during pupal development: the 1st day of pupation (1d), the 3rd day of pupation (3d), the 6th day of pupation (6d), and the 9th day of pupation (9d). As shown in Figure 3A, the expression level of *BmTMED6* in the ovaries is relatively low from the 1d to the 3d. Subsequently, there is a notable surge in *BmTMED6* expression on the 6d, reaching approximately 4000-fold compared to the 3d. However, by the 9d, the expression of *BmTMED6* decreases approximately 1000-fold compared to the 3d, indicating a significant alteration in *BmTMED6* expression in the ovary during pupal development. WB analysis further supports these findings, demonstrating a gradual increase in BmTMED6 protein levels in the ovaries throughout development (Figure 3B,C).

### 3.4. Effect of BmTMED6 Knockdown in Female Pupae

In connection with the above results, the expression pattern of *BmTMED6* is different between the sexes in the larval stage and significantly changes during the critical period of ovarian development. To investigate the biological function of the *BmTMED6* gene in females, we inject dsBmTMED6 (dsRNA targeting *BmTMED6*) into female pupae on the 1st day of the pupal stage. The control group receives an injection of dsEGFP instead of dsBmTMED6 under identical conditions. The efficacy of knockdown is confirmed by assessing mRNA levels using qRT-PCR 24 and 48 h after injection (Figure 4A). We observe the ovaries at 3d, 6d, and 9d and find no significant differences in the external morphology between the experiment group and the control group (Figure 4B). Additionally, there are no significant differences observed between the experimental and control groups in terms of pupal developmental time and survival rate. Furthermore, there are no significant differences observed in the mating success rate or the proportion of female moths capable of ovulation (Table 1). However, the number of eggs laid by female moths in the experimental group decreased significantly; the control group lays an average of 226 eggs, while the experimental group lays an average of only 141 eggs, a decrease of about 37.6% (Table 2). Dissection of the female moths after oviposition reveals that many eggs in the experimental group remained within the abdomen and were not laid. The unlaid eggs are primarily located in the upper portion of the ovarian tubes, with a few near the ovipositor in the experimental group. In comparison, the control group has fewer unlaid eggs, and the unlaid eggs are distributed throughout various sections of the ovarian tubes in the abdomen of the moths (Figure 4C).

### 3.5. The Impact of BmTMED6 on BmDop2R Expression

In most insects, it is typical to have a single subtype of *Dop2R*. However, in Lepidoptera, which included species such as *Plutella xylostella*, *Amyelois transitella*, and *B. mori*, two distinct subtypes of *Dop2R* were observed [19]. In previous studies, we found that inhibiting *BmDop2R* could reduce the number of laid eggs by female moths, which was similar to the effect observed when knocking down *BmTMED6*. We utilize the total RNA extracted within 24 h in Figure 4A to assess the expression levels of *BmDop2R1* and *BmDop2R2*. Compared to the control group, there is a significant decrease in the mRNA expression levels of *BmDop2R1* and *BmDop2R2* (Figure 5A). To further investigate the relationship between *BmTMED6* and *BmDop2R*, we overexpressed *BmTMED6* in BmN cells. As expected, when *BmTMED6* is transiently overexpressed, there is a significant increase in the mRNA expression levels of *BmDop2R1* and *BmDop2R2* (Figure 5B).

## 4. Discussion

TMED proteins have been demonstrated to influence the maturation of the reproductive system in *Caenorhabditis elegans* [20,21]. Similarly, certain TMED proteins in *D. melanogaster* (eclair, CG9308, CHOp24, p24-1, and logjam) exhibit unique sex-specific expression patterns and are linked to female reproduction [13,22]. TMED6 is expressed in multiple tissues in mice, with pancreatic cells displaying a distinct expression pattern [5,11]. TMED6 displays a unique expression pattern in the brain, ventral nerve cord, egg, and ovary in *D. melanogaster* [12,22]. Our previous study demonstrated the abundant expression of BmTMED6 across all stages and tissues of the silkworm, implying its significant involvement in the life activities of silkworms [6]. In this study, we investigate the gender expression pattern of BmTMED6 in eight larval tissues of silkworms. We observe differences between males and females, particularly in the head and fatbody. The gender-specific differential expression observed in the head suggests that BmTMED6′s influence on oviposition behavior may be related to the nervous system. Previous research indicates that TMED is involved in stress responses and acts as a regulatory factor in maintaining cellular homeostasis [14,23,24,25]. The fatbody in silkworms plays a crucial role in nutrient storage, immunity, detoxification, and various synthetic metabolic processes, similar to the liver in mammals. The specific expression of BmTMED6 in the fatbody suggests its involvement in gender differences related to these functions of the fatbody tissue. Furthermore, we identify a significant disparity in *BmTMED6* expression within the ovaries of pupae, with an approximately 4000-fold variation depending on the developmental stage. There are also significant differences in protein levels. Due to the structural similarity of TMED family proteins, there is non-specific band image noise in our WB results, which leads to errors in gray value analysis. Errors caused by TMED-specific antiserums are also present in *D. melanogaster* and mammals [13,26]. These strongly suggest a potential correlation between BmTMED6 and silkworm female reproduction.

*Logjam* (*DmTMED6*) mutations result in the blockage of egg laying and the retention of eggs in the reproductive duct of female *D. melanogaster* [12,15,17]. To explore the function of *BmTMED6*, we downregulate its expression in female pupae on the first day of pupation. The results demonstrate a significant reduction in the number of laid eggs and an increase in the number of eggs retained within the bodies of female moths. Our observation is similar to that of *D. melanogaster*. However, the total number of eggs produced by the moths and the development and morphology of the ovaries in the pupae do not display significant differences compared to the control group. Additionally, the unlaid eggs of the experimental group are primarily situated in the upper portion of the ovarian tubes, far from the ovipositor. After mating, the eggs that are lined in the ovarian tube of the female moth move sequentially towards the ovipositor by the peristalsis of the oviduct, and they are then expelled by the female moth [27,28]. Therefore, the decrease in the number of eggs laid may be due to weak peristalsis of the ovarian ducts in the female moths with dsBmTMED6. These findings indicate the role of *BmTMED6* as a regulator of the oviposition process in silkworms and other Lepidoptera.

Bioamines are important neurotransmitters and neurohormones [29]. Biogenic amine receptors, as part of the G protein-coupled receptor family, have been extensively investigated for their effects on female insect reproduction [30,31,32,33]. TMED proteins play a role in the transportation of G protein-coupled receptors and have been associated with neurodegenerative disease [4,34]. In *D. melanogaster*, it has been suggested that the *DmTMED6* gene regulates ovulatory behavior by influencing octopamine or octopamine receptors [13,14]. TMED6 has been found to affect insulin secretion in mammals, while *Dop2R* is also involved in mediating insulin secretion [11,35,36,37]. In our previous experiments, inhibition of *BmDop2R* resulted in a significant decrease in egg laying and an increase in the number of unlaid eggs by female moths, which is similar to that of knockdown *BmTMED6*. In our current study, we have discovered that *BmTMED6* is capable of influencing the expression of *BmDop2R*. These indicated that there was a close relationship between the expression of *BmDop2R* and *BmTMED6*. This is further supported by the differential expression of BmTMED6 in male and female silkworm heads, which contain abundant neural cells. So, *BmTMED6* likely affects moth oviposition by influencing *Dop2R*.

Egg laying is a crucial reproductive behavior in female insects, and any abnormalities in this process can have significant effects on their reproduction and population sustainability. Understanding the mechanisms that regulate female insect reproduction is a key focus of research in the fields of economic insect rearing and agricultural pest control. This study provides a new perspective on the regulation of reproduction in silkworms and other female insects. It establishes a foundation for future in-depth investigations into the specific mechanism by which BmTMED6 regulates female reproduction through Dop2 and for understanding the molecular mechanisms underlying the regulation of egg-laying in silkworms.

## 5. Conclusions

In conclusion, our study establishes BmTMED6 as a critical regulator in silkworm female reproduction, highlighting its potential application in sericulture and pest management.

## Figures and Tables

**Figure 1 insects-15-00103-f001:**
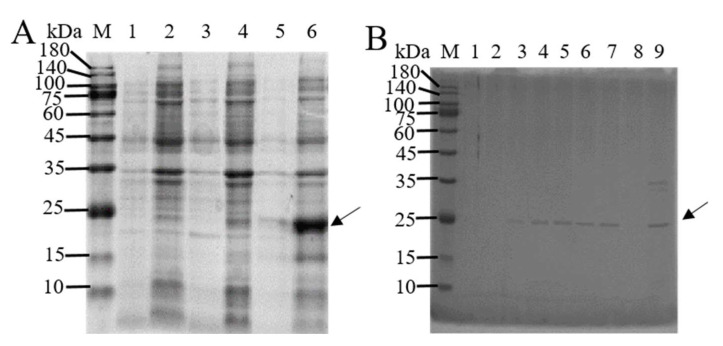
SDS-PAGE analysis of the BmTMED6 recombinant protein. (**A**) Induction of protein expression. M: Marker; 1: Supernatant at 16 °C; 2: Precipitate at 16 °C; 3: Supernatant at 28 °C; 4: Precipitate at 28 °C; 5: Supernatant induced at 37 °C; 6: Precipitate induced at 37 °C. The arrows indicate the BmTMED6 recombinant protein. (**B**) Elution results with different imidazole concentrations. M: Marker; 1–8: Elution flowthrough with 25 mM, 50 mM, 75 mM, 100 mM, 125 mM, 150 mM, 250 mM, and 500 mM imidazole, respectively; 9: protein flowthrough. The arrows indicate the BmTMED6 recombinant protein.

**Figure 2 insects-15-00103-f002:**
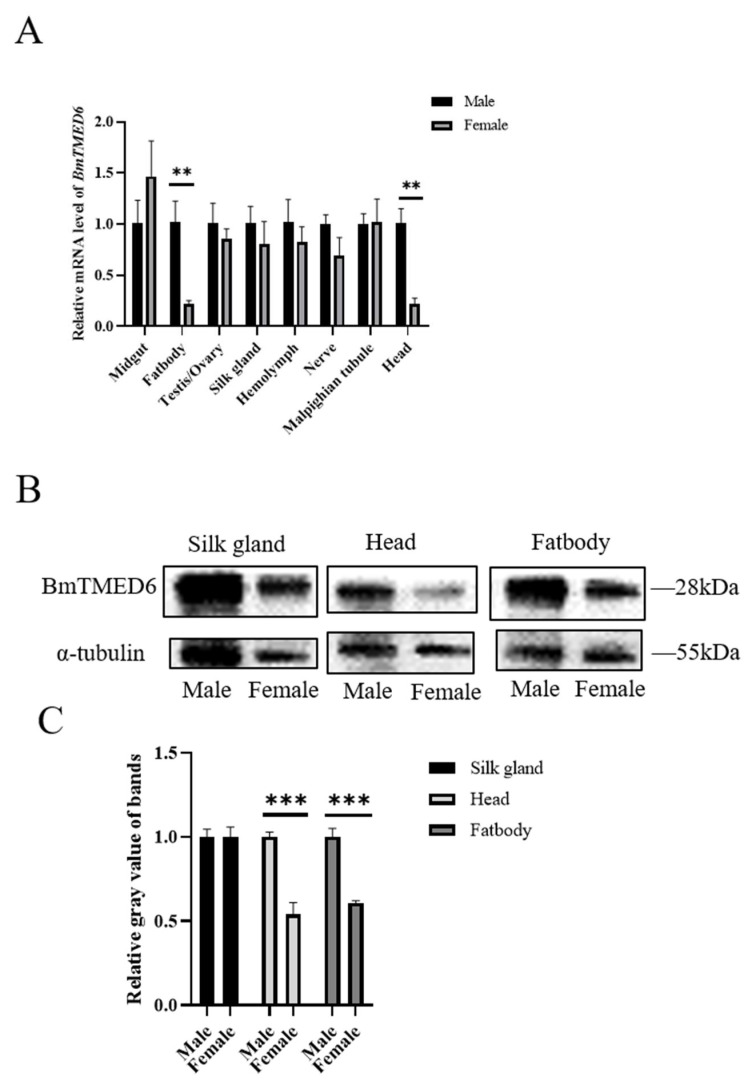
Expression of BmTMED6 in larval tissues on the 3rd day of the 5th instar in different sexes. (**A**) Relative mRNA expression levels of *BmTMED6* are assessed by qRT-PCR in eight tissues. ** *p* < 0.01. (**B**) Protein expression of BmTMED6 in three tissues. (**C**) Analysis of WB bands relative gray value of BmTMED6 in three tissues. *** *p* < 0.001.

**Figure 3 insects-15-00103-f003:**
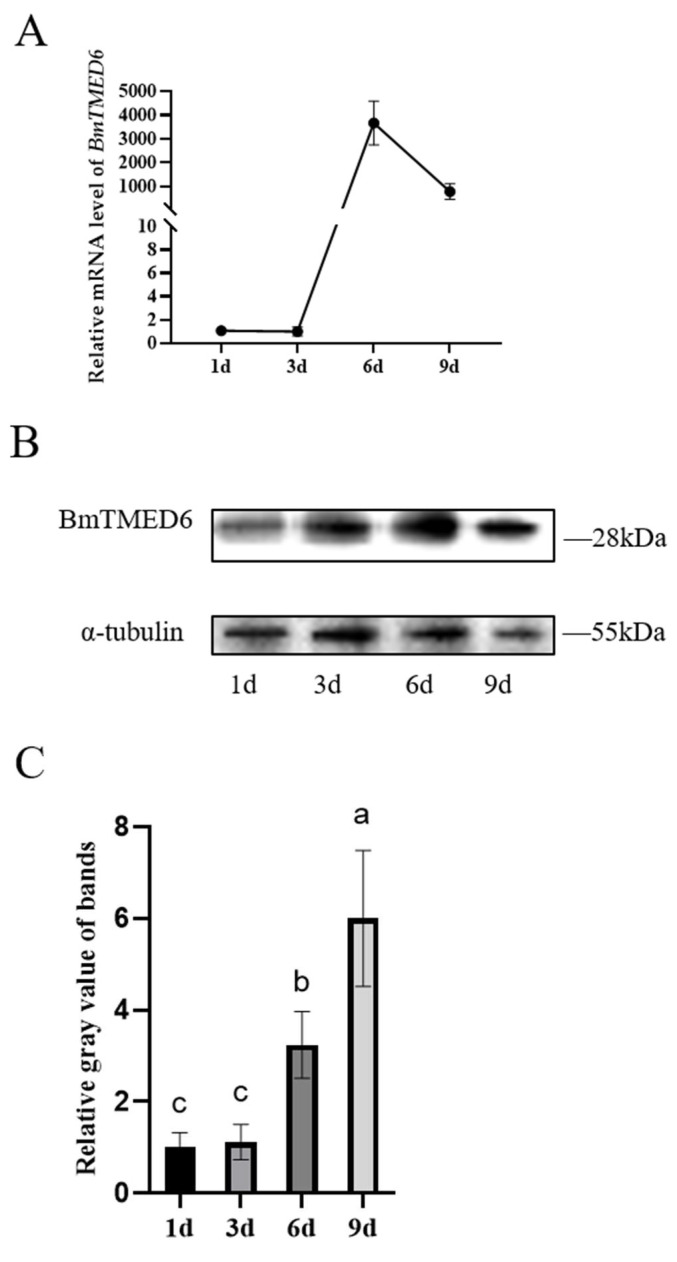
Expression levels of BmTMED6 in the ovaries of pupae. (**A**) Relative mRNA expression levels of *BmTMED6*. (**B**) Ovarian protein expression of BmTMED6 at different periods. (**C**) Analysis of relative gray value of WB bands of BmTMED6 in ovary, a–c represent the level of the mean from high to low, based on statistical significance.

**Figure 4 insects-15-00103-f004:**
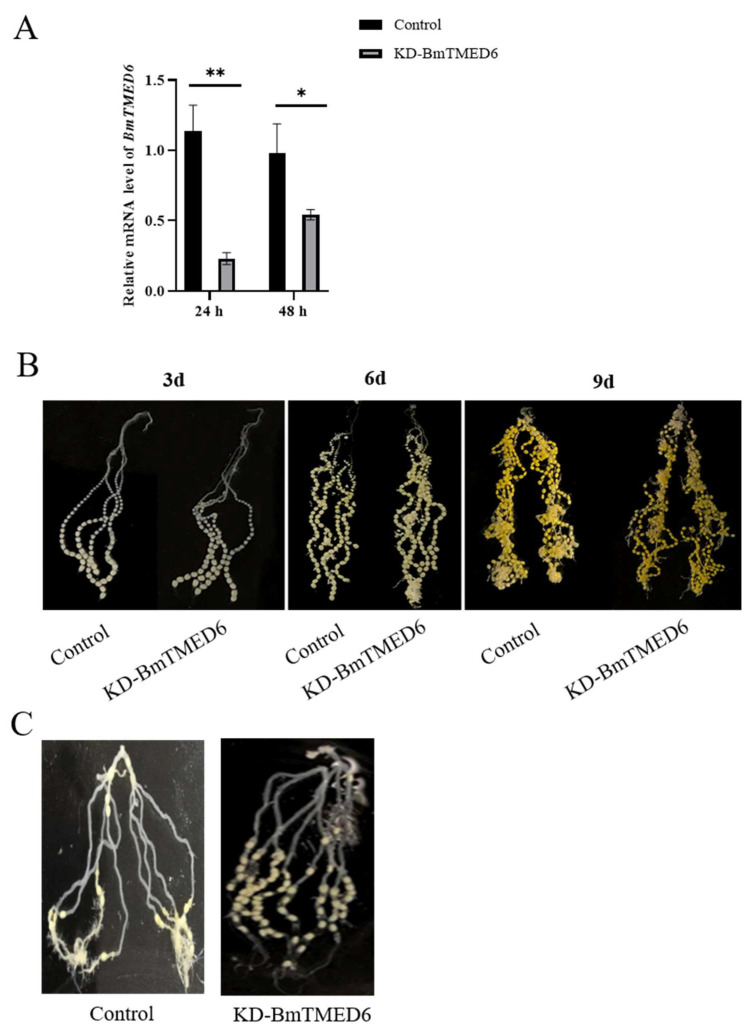
The impact of *BmTMED6* on female reproduction. KD-BmTMED6: Experimental group injected with dsBmTMED6. (**A**) Knockdown efficacy assessed by qRT-PCR at 24 h and 48 h post-injection. 24 h: 24 h after injection; 48 h: 48 h after injection; * *p* < 0.05, ** *p* < 0.01. (**B**) Ovarian morphology on the 3rd, 6th, and 9th days of the pupal stage. (**C**) Unlaid eggs in the ovaries after oviposition by female moths.

**Figure 5 insects-15-00103-f005:**
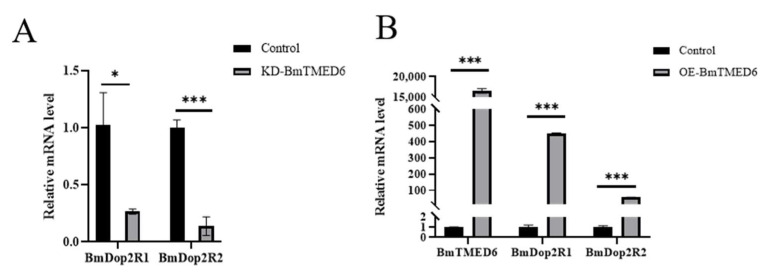
Expression levels of *BmDop2R1* and *BmDop2R2* are analyzed by qPCR. (**A**) Relative mRNA expression levels of BmDop2R1 and *BmDop2R2* in female pupae after *BmTMED6* knockdown. * *p* < 0.05, *** *p* < 0.001. (**B**) Relative mRNA expression of *BmDop2R1* and *BmDop2R2* when *BmTMED6* is transiently overexpressed 16,000 times in BmN cells. *** *p* < 0.001.

**Table 1 insects-15-00103-t001:** The development time and survival rate of pupae, the mating success rate of female moths, and the proportion of female moths able to ovulate.

Group	Development Time (d)	Survival Rate	Mating Success Rate	Proportion of Female Moths That Ovulate
Control	9.5	100%	100%	100%
KD-BmTMED6	9.5	100%	100%	100%

**Table 2 insects-15-00103-t002:** Comparison of laid eggs, unlaid eggs, and total egg production between control and experimental groups.

Group	Number of Laid Eggs	Number of Unlaid Eggs	Total Egg Production
Control	226.38 ± 32.513 a	14.92 ± 7.205 d	241.31 ± 32.778 a
KD-BmTMED6	141.00 ± 49.346 b	70.85 ± 21.385 c	211.85 ± 36.792 a

Note: a, b, c, and d represent the levels of the mean values from high to low, based on statistical significance.

## Data Availability

All the datasets in this study can be provided upon reasonable request.

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
