# Peer review of "The Role of BmTMED6 in Female Reproduction in Silkworm, Bombyx mori"

_insects, 2024, doi:10.3390/insects15020103_

Round 1
Reviewer 1 Report
Comments and Suggestions for Authors
In this short article, the authors investigate the function of TMED6 in silkworm reproduction. They show that during the larval stages, the expression of BmTMED6 is different in male and female, specifically in the fat body and the head. Additionally, by using RNAi approach, they show that BmTMED6 loss of function impacts the female eggs laying with an accumulation of unlaid eggs in the ovarian tubes. Finally, by manipulating BmTMED6 expression, they uncover changes in BmDop2R1 and BmDop2R2 expressions, suggesting that BmTMED6 plays a role in oviposition potentially through the dopamine signaling pathway.
Overall, the novelty of the paper is limited since it reproduces data already available in greater details on oviposition defects due to p24 loss of function in other insects. The final section regarding the BmTMED6 on BmDop2R expression is potentially interesting. However, its rationale is not clearly introduced and previous data are not presented or cited, the impact on male fecundity is not tested and the hypothesis is quickly tested in 2 sets of qPCR in different experimental setups (injected female silkworm for BmTMED6 knockdown and tissue culture for its overexpression)
In my opinion, the paper would need to be strengthened to warrant publication. The quality of the western blots and some images is low and make it difficult to understand. I would suggest the authors to strengthen their data in the final section by testing experimentally the dopamine signaling in oviposition in silkmoth or by clearly citing available data if they exist.
Specific points
The section 3.1 has no real interest and can be easily omitted. It does not provide any data regarding BmTMED6 since it is a step to generate an antibody.
The western blots in the figures 2B are not convincing. Particularly in the fat body lanes, the bands look irregular and can lead to misinterpretation. In the same line, the quality of the Western blots in Figure 3B is really low.
In the Fig 4C, I could not understand the 2 panels on the left. The quality of the images is really low and prevent appropriate interpretations. In addition, there are not clearly described in the figure legends.
Line 226 : the effect of BmDop2R loss of function in the oviposition rate of female moths is briefly mentioned. However, the data are not shown or the corresponding citation is not given. It is impossible for readers to appreciate whether the phenotypes are strictly similar as mentioned by the authors.
Line 260 : “Additionally, the unlaid eggs were primarily observed in the upper portion of the ovarian tubes, suggesting a lack of motivation to move toward the ovipositor for discharge.” The authors may want to consider rephrasing this sentence. What is a lack of motivation? Do they propose that females’ behaviors are changed upon BmTMED6 loss of function? This has to be clarified or demonstrated.
Minor comments:
Line 133: The 2.7 section header is similar to 2.6 and has to be changed.
Comments on the Quality of English Language
I do not have any problems with the quality of the English language.
Author Response
Dear reviewer,
Thank you for your valuable feedback and insightful comments on our manuscript. We have carefully addressed each of your suggestions and made significant revisions to the paper accordingly. Below, we provide responses to your comments:
Issue 1: Lack of data or relevant references on the effect of BmDop2R functional inhibition on moth oviposition.
Response: The data related to the functionality of BmDop2R are derived from our unpublished results. As the focus of this paper is primarily on the preliminary exploration of BmTMED6 functionality, to avoid duplication and maintain the paper's focus, we did not include the BmDop2R-related data in the manuscript. This response letter includes the unpublished data on the impact of inhibiting BmDop2R on oviposition as supplementary material.
Issue 2: Poor quality of protein blots and images in Figures 2B, 3B, and 4C.
Response: We have repeated the Western blot experiments, performed grayscale analysis on the new Western blot results, and included the updated data in the revised manuscript. However, due to the structural similarity within the TMED protein family, there may be interference and image noise in our bands. This observation has been mentioned in the revised Discussion section along with relevant information from other studies involving TMED proteins. We apologize for the low-quality representation in the first image of Figure 4C, which depicted the quantity of eggs laid. Considering the poor image quality, we have removed one image from Figure 4C. Specific details regarding egg quantity have been analyzed and listed in a table.
Issue 3: The phrase "lack of motility" may not fully express our intended meaning.
Response: Since unlaid eggs are deposited in the upper part of the oviduct, we speculate that the loss of BmTMED6 function may weaken the peristaltic movement in the oviduct of female moths. We have made this change in our revised version.
Other details have been modified or supplemented.
We have made the necessary changes and additions to address other details as well. Once again, we appreciate your comprehensive review and constructive suggestions. Your feedback is highly valuable in enhancing the clarity and quality of the manuscript. We hope this revised manuscript meets your expectations. If you have any further suggestions or concerns, please let us know.
Sincerely,
Chunyang Wang
College of Sericulture, Textile and Biomass Sciences, Southwest University,
Chongqing, China

Reviewer 2 Report
Comments and Suggestions for Authors
General comments
In the work of Wang and co-workers, the authors have studied the role of Transmembrane emp24 domain 6 protein (TMED6) in the oogenesis and oviposition of the silkworm, Bombyx mori. TMED proteins facilitate the transport and folding of specific proteins and are well studied in mammals but scarcely investigated in insects. In order to assess the function of TMED6 in B. mori females, the authors employed a combination of biochemical and cell biology approaches. The work deals with a relevant issue and intends to fulfil a gap in the knowledge of TEMD proteins in insects in general and in the silkworm in particular. The manuscript is well organized and its objectives as well as the experimental design are clear and straightforward. The findings are compelling and the conclusions match the reach of findings. Nevertheless, the are some point that need to be corrected, rewritten and/or expanded. Some suggestions are included below.
Specific comments
Simple summary:
- Page 1, lines 15-16: what does it mean that TEMD6 and dopamine D2-like receptors have a close relationship? A close relationship regarding what? That point is unclear. Please specify.
Abstract:
- Page 1, line 21: Scientific names such as Drosophila melanogaster should be italicized. Please correct.
Introduction:
- Page 2, line 50: What does "each insect" mean? Did the authors mean "each insect order investigated"? Please rewrite.
- Page 2, line 51: That sentence is not clear. Please reword it.
- Page 2, line 52: Consider replacing "the report was abundant in mammals" by "the reports are abundant in mammals".
- Page 2, lines 54-55: Seems to be a mistake in "colorectal cancer, cancer and cancer". Please correct.
- Page 2, lines 59-60: The authors said that "there have been no reports on the TMED function" regarding silkworms but in the next sentence they imply that they have previous results in that topic. They also have a reference in that regard: WANG, C.; GUO, Y.; LI, H.; Chen, P. Analyzing the evolution of insect TMED gene and the expression pattern of silkworm 318 TMED gene. Chinese Journal of Biotechnology 2023, 1-15, doi:10.13345/j.cjb.230251. Please clarify this point.
- Page 2, lines 64-66: I do not see how the expression of a gene involved in oviposition has significant implication in the development of insect management strategies. Please expand.
Materials and methods
- Page 3, lines 113-114: Is correct that the authors loaded 80 mg of protein in each well? If so, what type of electrophoretic setting was employed? No normal electrophoretic setting can handle such an amount of protein.
- Page 3, line 133: Page 3, line 133: The title of the section is not correct.
Results
- Figures 2B and 3B are of very low quality. Please improve and add a densitometric analysis of the Western blots.
- Page 4, lines 167-170: qPCR results do not necessarily have to match those of Western blot, that is why I do not agree with using the expression “To validate the reliability of the qRT-PCR findings…”. Please rephrase. It is not clear to me why the authors select those 3 organs to perform Western blot and not others. That choice needs to be justified.
- A densitometric analysis of the Western blot of Fig. should be included.
- Page 6, lines 193-197: That paragraph presents a different font. Please homogenize.
- Why did the authors choose to focus on the ovary after screening 8 different organs (Fig. 2)? That justification is lacking.
- Page 6, lines 202-203: The expression “To confirm the results, we observed 10 individuals in each group.” Is confusing. What was the number of individuals observed for the results presented in Figure 4A and B? Please clarify.
- Methodology regarding Fig. 5B is not sufficiently described. Please complete.
Discussion
- Page 8, line 249: Once the complete scientific name is written for the first time in full, the second time and thereafter only the initial of the genus must be written (e.g. D. melanogaster). Please check and correct, throughout the manuscript.
- Page 8, lines 249-251: Please explain in more detail the differences between the work conducted in reference 6 and the present work. Since I could not access the reference number 6 I was not able to compare what were the actual differences.
- Page 8, lines 251-252: Again, it sounds odd the fact that the authors found differences in the expression of BmTMED6 in the head and the fat body when they compared males and females, but they study further the expression in the ovary.
- Page 8, lines 262-263: I suggest rephrasing “lack of motivation” since it does not apply to this situation.
- Page 9, lines 265-267: It seems that the authors measured the developmental progress and survival rate of pupae with dsBmTMED6. Please include those results in the manuscript.
- Did the authors perform an experiment directly using dopamine (by injecting it in pupae, for example) trying to rescue the oviposition phenotype and/or evaluating the consequences of such treatment on BmTMED6 expression?
Comments on the Quality of English LanguageShould be improved.
Author Response
Dear reviewer,
Thank you for your thoughtful comments and suggestions. We have taken each point into careful consideration and made substantial revisions accordingly. Here is our response to your feedback:
- Page 1, lines 15-16: We have made the following additions: BmTMED6 is closely associated with the dopamine receptor, suggesting its potential involvement in regulating the oviposition behavior of the moth through dopamine-related pathways.
- Page 2, lines 59-60: We have made the necessary changes, and currently, we have published one report on BmTMED in silkworms. This study represents the second article on BmTMED in silkworms. We have clarified this point in the revised manuscript.
- Page 2, lines 64-66: The significance of TMED6 in the development of insect management strategies has been expanded upon in the Discussion section to provide a clearer explanation.
- Page 3, lines 113-114: We apologize for the error in mentioning the protein loading amount. It was indeed a mistake. The correct protein loading amount will be provided in the revised manuscript.
- We have repeated the Western blot experiments, performed grayscale analysis on the new Western blot results, and included the updated data in the revised manuscript. However, due to the structural similarity within the TMED protein family, there may be interference and image noise in our bands. This observation has been mentioned in the revised Discussion section along with relevant information from other studies involving TMED proteins.
- The rationale for selecting three organs for Western blot analysis is as follows: The fifth instar stage of silkworms is a critical period for silk gland development, and qRT-PCR results have shown differential expression patterns in the fatbody and head. Our objective is to investigate the protein expression levels in the silk gland during this crucial stage using Western blot analysis and validate the relative expression levels in the fatbody and head.
- The reason for focusing on the ovaries is as follows: In the fruit fly, DmTMED6 serves as a key factor in regulating female oviposition behavior. In silkworms, the pupal stage represents a critical period for the development of female moth ovaries. We observe gender-specific expression patterns in silkworms and are interested in investigating whether BmTMED6 is also involved in female reproduction in silkworms. Therefore, we examine the developmental trend of BmTMED6 in the ovaries of female pupae.
- Page 6, lines 202-203: We apologize for the confusion caused. We have made changes and provided detailed supplements in the revised manuscript to address this issue.
- Regarding Figure 5B, we will provide a more detailed description of it in the revised manuscript.
- Page 8, lines 249-251: In reference to [6], we identified the TMED family genes of silkworm, Tribolium castaneum, tobacco moth and Italian bee from their genomes, and found that the TMED family gene composition patterns of one α-class, one β-class, one δ-class and several γ-classes arose in the common ancestor of pre-divergent Hymenoptera insects, while the composition of Drosophila TMED family members has evolved in a unique pattern. Insect TMED family γ-class genes have evolved rapidly, diverging into three separate subclasses, TMED6-like, TMED5-like and TMED3-like. The TMED5-like gene was lost in Hymenoptera, duplicated in the ancestors of Lepidoptera and duplicated in Drosophila. Insect TMED protein not only has typical structural characteristics of TMED, but also has obvious signal peptide. There are seven TMED genes in silkworm, distributed in six chromosomes. One of seven is single exon and others are multi-exons. The complete open reading frame (ORF) sequences of seven TMED genes of silkworm were cloned from larval tissues and registered in GenBank database. BmTMED1, BmTMED2 and BmTMED6 were expressed in all stages and tissues of the silkworm, and all genes were expressed in the 4th and 5th instar and silk gland of the silkworm. The present study revealed the composition pattern of TMED family members, their γ class differentiation and their evolutionary history, providing a basis for further studies on TMED genes in silkworm and other insects. We use the past tense to describe our prior research and the methods in the current paper, while other studies are described in the present tense to distinguish them.
- Page 8, lines 262-263: The phrase "lack of motility" may not fully express our intended meaning. Since unlaid eggs are deposited in the upper part of the oviduct, we speculate that the loss of BmTMED6 function may weaken the peristaltic movement in the oviduct of female moths. We have made this change in our revised version.
- Page 9, lines 265-267: We have included additional results related to pupal developmental time and survival rate.
- The data related to BmDop2R are derived from our unpublished results. The dopamine rescue experiments are associated with dopamine receptor functionality. The focus of this paper is to investigate the function of BmTMED6. In order to avoid duplicate publication and maintain the research emphasis, we have not included the relevant data in the manuscript.
- Other relevant suggestions have been addressed and incorporated into the revised manuscript.
Thank you once again for thoroughly reviewing our paper and providing valuable suggestions. Your feedback is highly appreciated in improving the clarity and quality of our manuscript. If you have any other suggestions or concerns, please feel free to let us know.
Sincerely,
Chunyang Wang
College of Sericulture, Textile and Biomass Sciences, Southwest University,
Chongqing, China
Reviewer 3 Report
Comments and Suggestions for Authors
The MS named"The Role of BmTMED6 in Female Reproduction in Silkworm, Bombyx mori." It focuses on the study of TMED6, a transmembrane emp24 domain protein, and its role in female reproduction in silkworms. The research investigates how TMED6 expression in various tissues influences the number of eggs laid by female moths and the relationship between TMED6 and dopamine D2-like receptors. The findings are significant for understanding the molecular mechanisms underlying female reproduction in silkworms and have implications for sericulture production and insect management strategies.
After reviewing the manuscript, I found that it is well-structured and presents its research clearly. However, there are a few areas where language improvements could enhance the clarity and readability of the paper:
Minor
Abstract:Suggestion: Clarify the sentence structure in certain parts to make it more concise. For example, in line 25, consider rephrasing to avoid the passive voice.
Example: "We found that reduced TMED6 expression significantly decreases the number of eggs laid by female moths, leading to an accumulation of unlaid eggs in the abdomen."
Introduction: Suggestion: Ensure consistency in tense usage. The introduction section fluctuates between past and present tense, which can be confusing for readers.
Example: Maintain a consistent use of past tense for previous studies and present tense for the current study's aims and methods.
Materials and Methods:
Suggestion: Provide more detailed descriptions in the methodology to enhance reproducibility. Some procedures are mentioned briefly and could benefit from more specifics.
Clarify the sample size and selection criteria in the experiments. It's essential to know how many silkworms were used and how they were chosen to ensure the validity of the results.
The title of method 2.7 is incorrect, and it used GraphPad for data analysis. It is suggested to use SPSS instead.
Results Some sentences are quite lengthy and could be broken down for clarity. Divide the long sentence into two for better readability.
Discussion Suggestion: Avoid redundancy in discussing the findings and their implications.
Example: Streamline the discussion of TMED6's role in female reproduction to avoid repeating points already made in the results section.
Expand on the implications of these findings for sericulture and insect management. The current discussion is limited and could be broadened to include potential applications or future research directions.
Conclusion The conclusion could be more impactful by succinctly summarizing the key findings and their broader implications.
Example: "In conclusion, our study establishes BmTMED6 as a critical regulator in silkworm female reproduction, highlighting its potential application in sericulture and pest management."
Comments on the Quality of English Language
Here are a few areas where language improvements could enhance the clarity and readability of the paper.
Author Response
Dear Reviewer,
Thank you for providing valuable feedback and insightful suggestions on our manuscript. We have made significant revisions according to your advice. Below, we will respond to your comments point by point:
Issue 1: Clarify the sentence structure in certain parts to make it more concise.
Response: We have revised sentence structures to avoid using passive voice.
Issue 2: Suggestion: Ensure consistency in tense usage. The introduction section fluctuates between past and present tense, which can be confusing for readers.
Response: In the revised draft, we will use the past tense to describe our previous research and the methods in this paper, while using the present tense to describe this study and relevant research in other animals, ensuring consistency.
Issue 3: Suggestion: Provide more detailed descriptions in the methodology to enhance reproducibility.
Response: We will provide more detailed method descriptions in the revised draft to enhance reproducibility. We will clearly state the sample size and selection criteria.
Issue 4: The title of method 2.7 is incorrect, and it used GraphPad for data analysis. It is suggested to use SPSS instead.
Response: We have made the changes.
Issue 5: Results Some sentences are quite lengthy and could be broken down for clarity. Divide the long sentence into two for better readability.
Response: You mention that some sentences are too long and can be split into two for clarity. In the revised draft, we will split long sentences into two to improve readability.
Issue 6: Discussion Suggestion: Avoid redundancy in discussing the findings and their implications.
Response: We will simplify the discussion of TMED6's role in female reproduction to avoid repeating points already mentioned in the Results section.
Issue 7: Expand on the implications of these findings for sericulture and insect management. The current discussion is limited and could be broadened to include potential applications or future research directions.
Response: The significance of TMED6 in silkworm rearing and insect management has been expanded in the discussion to provide a clearer explanation.
Issue 8: Conclusion The conclusion could be more impactful by succinctly summarizing the key findings and their broader implications.
Response: The conclusion section has been revised.
Other details have been modified or supplemented.
Your feedback is highly valuable in improving the clarity and quality of our manuscript. We hope this revised manuscript meets your expectations. If you have any further suggestions or concerns, please let us know.
Sincerely,
Chunyang Wang
College of Sericulture, Textile and Biomass Sciences, Southwest University,
Chongqing, China
Reviewer 4 Report
Comments and Suggestions for Authors
The present study investigated the function of BmTMED6 in egg production by female silkworm, and its relationship with BmDop2R1 and BmDop2R2.
The present research is straightforward and built on previous research from the lab. It reported the phenotype of BmTEMD6 knock down in female silkworm. However, it could have been better if the authors looked into the underlying mechanisms of the phenotype.
In the Introduction session, please give more background information on Dop2R.
In the Method session, session 2.3, please describe how the protein in inclusion bodies was purified.
Line 126 and 133 are the same. Please correct it.
In the Results session, session 3.1 should be in supplemental, as it only describes the purification of protein and the antibody generation. Again, what’s the method used to purified inclusion body protein?
The font is not consistent throughout the Results session. Please unify it.
Figure 2B, the difference in the fatbody blot between male and female does not look as dramatic as shown in the mRNA level. Please provide quantification of the western blots by the band intensity. If the quantification is not consistent with mRNA level. Please provide discussion in possible reasons.
Please enlarge the first two panels in figure 4 C. It’s very difficult to see any significant difference based on the images provided.
In the discussion session, please discuss the potential mechanism underlying the impact of BmTMED6 and the dopamine pathway on the motivation and coordination of oviposition. Also, please discuss your future direction based on this current study.
Comments on the Quality of English LanguageThe quality of English is fine.
Author Response
Dear reviewer,
Thank you for your valuable feedback and insightful comments on our manuscript. We have carefully addressed each of your suggestions and made significant revisions to the paper accordingly. Below, we provide responses to your comments:
Issue 1: In the Introduction session, please give more background information on Dop2R.
Response: We appreciate your suggestion regarding providing additional background information on Dop2R in the introduction. However, to maintain the focus of this paper on the preliminary exploration of BmTMED6 function and ensure readability, we have decided to present the background information on Dop2R in the discussion section.
Issue 2: In the Method session, session 2.3, please describe how the protein in inclusion bodies was purified.
Response: In section 2.3, we provide a detailed description of how to purify the protein from inclusion bodies.
Issue 3: Line 126 and 133 are the same. Please correct it.
Response: We apologize for the repetition in lines 126 and 133 and will correct it in the revised manuscript.
Issue 4: In the Results session, session 3.1 should be in supplemental, as it only describes the purification of protein and the antibody generation. Again, what’s the method used to purified inclusion body protein?
Response: In section 3.1, we provide a detailed description of the methods used to purify the protein from inclusion bodies.
Issue 4: Figure 2B, the difference in the fatbody blot between male and female does not look as dramatic as shown in the mRNA level. Please provide quantification of the western blots by the band intensity. If the quantification is not consistent with mRNA level. Please provide discussion in possible reasons. Please enlarge the first two panels in figure 4 C. It’s very difficult to see any significant difference based on the images provided.
Response: We have repeated the Western blot experiments, performed grayscale analysis on the new Western blot results, and included the updated data in the revised manuscript. However, due to the structural similarity within the TMED protein family, there may be interference and image noise in our bands. This observation has been mentioned in the revised Discussion section along with relevant information from other studies involving TMED proteins. We apologize for the low-quality representation in the first image of Figure 4C, which depicted the quantity of eggs laid. Considering the poor image quality, we have removed one image from Figure 4C. Specific details regarding egg quantity have been analyzed and listed in a table.
Issue 5: In the discussion session, please discuss the potential mechanism underlying the impact of BmTMED6 and the dopamine pathway on the motivation and coordination of oviposition. Also, please discuss your future direction based on this current study.
Response: We have added to the discussion in the new manuscript.
Other details have been modified or supplemented.
Once again, we appreciate your comprehensive review and constructive suggestions. Your feedback is highly valuable in enhancing the clarity and quality of the manuscript. We hope this revised manuscript meets your expectations. If you have any further suggestions or concerns, please let us know.
Sincerely,
Chunyang Wang
College of Sericulture, Textile and Biomass Sciences, Southwest University,
Chongqing, China
Round 2
Reviewer 2 Report
Comments and Suggestions for Authors
The authors have addressed most of the original comments. Nevertheless, some are still missing:
- Point 1 of the rebuttal letter: The authors claim that they added a sentence that is not present in the new version of the manuscript. The new version of that part does not solve what was originally requested. Please, add the sentence “BmTMED6 is closely associated with the dopamine receptor, suggesting its potential involvement in regulating the oviposition behavior of the moth through dopamine-related pathways.”.
- The original point “- Page 2, line 51: That sentence is not clear. Please reword it.” was not addressed nor rebutted. Please correct.
- The original point “- Page 2, line 50: What does "each insect" mean? Did the authors mean "each insect order investigated"? Please rewrite.” was not correctly addressed. The authors are discussing insect orders but then they mentioned “The four insect species retained a TMED6-like member.” (page 2, lines 49-50). Which insect species? Did the authors mean insect orders?
- Point 3 of the rebuttal letter: The fact that this point was expanded later does not solve the original point which was “Page 2, lines 64-66: I do not see how the expression of a gene involved in oviposition has significant implication in the development of insect management strategies. Please expand.”. Either explain it here or remove that part.
Comments on the Quality of English LanguageNo comments.
Author Response
Dear Reviewer,
Thank you for your patience, and we apologize for any confusion caused. We have made revisions based on your feedback:
1. We have added the following sentence as requested: "BmTMED6 is closely associated with the dopamine receptor, suggesting its potential involvement in regulating the oviposition behavior of the moth through dopamine-related pathways."
2. We apologize for not addressing the clarity issue with the sentence on page 2, line 51. We have reworded it as follows: "The slower evolutionary rate of the γ subfamily in insects compared to mammals, combined with the presence of a TMED6-like member in these insect orders, underscores the conserved functional importance of TMED6 in insects."
3. We have revised the sentence on page 2, line 50 to accurately reflect the investigated insect orders: "The orders Lepidoptera, Diptera, Coleoptera, and Hymenoptera all retained the TMED6-like member."
4. Finally, we have addressed the concern about the development of insect management strategies by removing the relevant statement from the manuscript.
Thank you for bringing these points to our attention, and we appreciate your assistance in improving the clarity and accuracy of our manuscript.
Sincerely,
WANG